# Exploring the Reciprocal Relationships between Happiness and Life Satisfaction of Working Adults—Evidence from Abu Dhabi

**DOI:** 10.3390/ijerph19063575

**Published:** 2022-03-17

**Authors:** Masood A. Badri, Mugheer Alkhaili, Hamad Aldhaheri, Guang Yang, Muna Albahar, Asma Alrashdi

**Affiliations:** 1Chairman Office, Department of Community Development, Abu Dhabi P.O. Box 30039, United Arab Emirates; mugheer@addcd.gov.ae (M.A.); muna.albahar@addcd.gov.ae (M.A.); 2College of Business and Economics, United Arab Emirates University, Al Ain P.O. Box 15551, United Arab Emirates; 3Undersecretary Office, Department of Community Development, Abu Dhabi P.O. Box 30039, United Arab Emirates; hamad.aldhaheri@addcd.gov.ae; 4Social Monitoring and Innovation Sector, Department of Community Development, Abu Dhabi P.O. Box 30039, United Arab Emirates; guang.yang@addcd.gov.ae (G.Y.); asma.alrashdi@addcd.gov.ae (A.A.); 5College of Humanities and Social Sciences, United Arab Emirates University, Al Ain P.O. Box 15551, United Arab Emirates

**Keywords:** life satisfaction, happiness, well-being, working adults, path analysis, reciprocal relation, Abu Dhabi

## Abstract

This paper examines the relationships between a range of well-being factors and two commonly used subjective well-being measures—happiness and life satisfaction. Data from the second cycle of the Quality of Life (QoL) Survey in Abu Dhabi were used, which included 32,087 working adults. The well-being factors included in the analysis covered various aspects of life themes: income and jobs, work–home balance, health and physical activities, social and community services, living environment, and family/friends’ relationships and connections. Using standardized data, path analysis yielded an optimal path model that suggested the presence of a reciprocal relationship between happiness and life satisfaction. In addition, the final model suggested that four variables—job satisfaction, mental health, satisfaction with relationships with people, and the size of the social support network—had direct effects on happiness and life satisfaction. The model also identified three variables—satisfaction with family life, mental health, and job satisfaction—to have the most significant effect on happiness.

## 1. Introduction

Well-being- or quality-of-life-related research in the context of Abu Dhabi has attracted much attention since the Department of Community Development launched the Abu Dhabi Quality of Life (QoL) Survey in 2019. Many public agencies in Abu Dhabi have used the QoL survey outcomes to form well-being indicators, aiming to develop social capital and promote happiness and life satisfaction among members of the communities. In this regard, several studies have emerged focusing on various aspects of quality of life among different segments of the population in Abu Dhabi, including the happiness of the elderly [1], self-perceived depression among adolescents [2], as well as social trust [3]. Badri et al. [4], in particular, addressed the effect of working time on workers’ quality of life. The proposed path model justified the significance of working hours on several well-being variables, implying that further research should include life satisfaction and happiness to address a more comprehensive model of quality of life for working people in Abu Dhabi.

Most research views life satisfaction as more complex than happiness. Nevertheless, the term is sometimes used interchangeably with happiness or well-being. Most see life satisfaction as the evaluation of one’s life. As a result, it is not simply one’s current level of happiness. Research views happiness as more “immediate, in-the-moment experience; although enjoyable, it is ultimately fleeting” [5]. As a result, a healthy life surely includes moments of happiness, but happiness by itself usually might not make for a fulfilling and satisfying life [6]. Research also identifies life satisfaction as being more “stable and long-lived” than happiness and broader in scope [7]. Life satisfaction usually reflects our general feeling about our life and how pleased we are with how it is going [8]. Research has explained that there are multidimensional factors that contribute to life satisfaction. Such domains might include work, relationships, relationships with family and friends, personal development, health, and other factors [9].

In the extant literature, well-being measures associated with working adults vary in the scope or domain of those studies and are often examined in different contexts [1,4]. Nevertheless, most research focuses on two subjective measures of well-being—life satisfaction and happiness—and explores some of the inter-relationships [1,10]. However, there has been little attention towards examining the possibility of reciprocal relationships between the more extended domain of life satisfaction and the narrower domain of happiness.

This current study of working people in Abu Dhabi is set to examine the possible indication of such reciprocal relations through an integrated path analysis approach. The study also aims to investigate the interactions between other well-being factors and life satisfaction and happiness. Such analysis is essential since it provides policymakers with a better understanding of the interactive relationships between happiness and life satisfaction, as well as more clarity regarding the interrelations of the different well-being constructs.

## 2. Review of the Literature

Researchers used to adopt objective well-being measures to evaluate people’s quality of life. In recent years, however, most quality-of-life projects in many countries have increasingly included both objective and subjective dimensions [11,12]. The OECD has used subjective well-being extensively, as it corresponds to how people experience and evaluate their lives and specific domains and activities in their lives [13]. Diener et al. [14] has provided a detailed review of scientific research on subjective well-being. The authors define subjective well-being to consist of a person’s cognitive and affective evaluations of his or her life. After providing a brief historical review of research on subjective well-being, they summarized the main measurement issues (e.g., the validity of self-reports, memory bias). In addition, they presented the major theoretical approaches to this area of research. In their research, they reviewed current findings on subjective well-being, and suggested future directions for the study of subjective well-being.

Life satisfaction is often considered as a subjective well-being measure that reflects positive and negative emotions [15,16]. Various studies examine overall or holistic life satisfaction or satisfaction from specific life domains, including family life, friend relationships, work and earnings, and schooling and education [17,18,19,20]. Some researchers add that life satisfaction is also a reflection of mental and physical health [21].

Research reveals many variables to be strongly associated with life satisfaction. Those variables are general well-being measures encompassing individuals’ satisfaction with income, jobs, employment, health, living conditions, social relationships, and connections [22,23,24,25]. International empirical studies broadly tend to comprehend that well-being is primarily affected by socioeconomic status and a healthy lifestyle [26,27]. Many studies report a strong relationship between life satisfaction and physical and psychological health [28,29,30,31]. Notably, some studies reveal stronger effects of the quality of connectedness and social relationships and a relatively less important influence of income on life satisfaction [17,32,33].

Defined as “a state of well-being and contentment” or “a pleasurable or satisfying experience” [11], happiness, as another commonly used subjective well-being measure, reflects the degree to which an individual perceives that his/her aspiration is met [27,28,29,30,31]. As happiness depends on both cognitive and emotional components, it reflects both psychological happiness and prudential happiness [11,32]. Valois et al. [15], and Nemati and Maralan [33] addressed happiness as a multi-dimensional construct made up of several significant factors of emotional, social relations with others, cognitive aspects, physical activities, and optimism. Likewise, Cloninger and Zohar [34], and Saricam [11] proposed that happiness may arise from a combination of factors, including other positive emotions and subjective well-being.

The relationship between life satisfaction and happiness is an important area within the field of positive psychology. Most researchers assert a positive link between life satisfaction and happiness [17,32,35,36,37], while seeing happiness as being more emotional and life satisfaction more cognitive in nature. For example, Myers [38] focused on happiness to explain positive experimentations when it comes to life satisfaction. Peterson et al. [39] argued that individual orientations to happiness predict life satisfaction. Nemati and Maralani [33] investigated the relationship between happiness and life satisfaction with the existence of some mediating factors related to resiliency. A study examining the structural relationships of well-being, leisure satisfaction, life satisfaction, and happiness concluded that life satisfaction and leisure play a major role in the perception of personal happiness [10]. The happiness achieved in a recreational activity plays a vital role in a person’s life satisfaction level [40]. Similarly, Liang et al. [41] pointed to happiness as a significant predictor of satisfaction with life, leisure satisfaction, national well-being, and personal well-being. Again, variables related to the composition of social networks and support from family and friends tend to receive the most attention concerning both life satisfaction and happiness [42,43].

While some analysts consider and use happiness interchangeably with life satisfaction [16], some other studies, nevertheless, focus on the abstract differences between happiness and life satisfaction [44,45]. Kahneman and Riis [46], and Kahneman et al. [47], for example, noted that happiness and satisfaction are distinct constructs, as happiness is a momentary experience that arises spontaneously, while life satisfaction is a long-term feeling based on achieving life-long goals. Seligman [48] also provided empirical evidence to highlight the conceptual difference between life satisfaction and happiness. Some researchers further explored the differences between life satisfaction and happiness through the effects from other factors such as positive mental health [49,50,51].

The happiness and life satisfaction of working adults have received much attention in recent years. Employee happiness or well-being is an emerging topic in management as well as in psychology [52]. Graham and Pettinato [53] posited economic conditions and jobs to be significant in determining happiness and proposed that a higher income, higher attainment of education, and job satisfaction have a noticeable influence on happiness. Ball and Chernova [19] reported the importance of job and income in working people’s well-being, life satisfaction, and happiness. However, some non-economic dimensions such as the quality of social relationships appeared to have a more significant impact on happiness and life satisfaction than income [54]. Therefore, some researchers concentrated on the importance of work–life balance and its influence on mental health, well-being, and happiness [27,55]. Erdogan et al. [56] also examined the influence of specific work-related factors on employee happiness and concluded that, amongst other factors, job satisfaction has highly significant effects.

The quality-of-life literature often reports bidirectional or reciprocal relationships between certain quality-of-life variables. Consistent with Keon and McDonald [57], Judge and Watanabe [58] found a reciprocal relationship between job and life satisfaction. Lu et al. [59] investigated the reciprocal relationship between psychosocial work stress and quality of life, factoring in the effects of gender and education. Unanue et al. [60] investigated the reciprocal relationship between gratitude and life satisfaction. Likewise, there are studies focusing on exploring the reciprocal relationship between material wealth and health [61]. However, few empirical works exist that examine such relationships between life satisfaction and happiness. Further research is also required to address the more complex relationships between happiness and life satisfaction as they relate to other well-being variables [23].

The objective of this current study is to explore the viability of a model encompassing life satisfaction, happiness, and other quality-of-life determinants for better understanding the direct and indirect relationships between these variables for working adults in Abu Dhabi.

## 3. Methods and Design

Based on the literature review, the direction of various paths in the model was envisioned and several variables were selected from the Abu Dhabi QoL Survey to test it. Furthermore, the model and analysis investigated both directional and reciprocal relations, especially concerning happiness and life satisfaction.

### 3.1. Participants

The study relied on the Abu Dhabi QoL Survey conducted in 2019/2020, which covered more than 72,000 respondents. A total of 34,499 were employed or self-employed and they constituted the target of this study. The online survey was available for Emiratis and all major non-Emirati community members. It was administered in six different languages (Arabic, English, Hindi (four different dialects), Farsi, Tagalog, and Chinese). Informed consent was obtained from all respondents involved in the study. The survey covered both full-time employment and part-time employment.

### 3.2. Instruments and Procedure

The design of the QoL survey was based mainly on several international well-being frameworks, including the OECD’s Better Life Index [13], World Happiness Report [62], Gallup Global Well-Being Survey [63], and European Quality of Life Surveys [64]. Conducted online, the QoL survey covered dimensions of housing, household income, jobs and earnings, health, education, safety, and social connections. Participation in the survey was voluntary and included all working individuals connected through databases provided by various government departments and private organizations in Abu Dhabi. Through an introductory letter, participants were informed about the objectives of the study and guaranteed confidentiality and anonymity. Both the Department of Community Development and the Abu Dhabi Statistics Center provided the ethical approval for this study.

The main variables from the survey that were identified for the current study included the subjective questions regarding overall job satisfaction, work–family balance, self-perception of health, mental health and depression, size of social support network, satisfaction with relations with other people, feelings about the surrounding environment, amount of quality family time, satisfaction with family life, frequency of meeting with friends, and life satisfaction and happiness.

For the variables used in the analysis, Table 1 provides further explanations. The table identifies the variables used, the definitions of each variable, and the corresponding scale. Since the scales are not unified, further standardization was performed before conducting the path analysis. The scales used (1–5) or (0–10) are consistent with the scales used by the countries in their annual Better Life Quality surveys [15].

### 3.3. Analysis Method

Relationships between happiness, life satisfaction, and other selected variables were tested using path analysis. The path model specified was estimated with the program LISREL8 using the Maximum Likelihood estimation procedure [65], which also provides a chi-square test of the models hypothesized. The methodology takes a confirmatory approach to the analysis [66]. The goodness of fit statistics were utilized to determine the adequacy of the model and whether the hypothesized relationships were supported and plausible. Specifically, the Normed Fit Index (NFI), Non-Normed Fit Index (NNFI), Comparative Fit Index (CFI), Goodness of Fit Index (GFI), Adjusted Goodness of Fit Index (AGFI), Root Mean Square Residual (RMR), and Standardized RMR were used. For these indices, values greater than 0.90 are typically considered to be acceptable. Values greater than 0.95 indicate a good fit to the data [66]. Moreover, values for (χ2/df) are considered satisfactory when <3. An RMSEA in the range of 0.05 to 0.10 is considered an indication of fair fit, while values smaller that 0.05 are considered a good fit [66].

The covariance matrix was also used when testing the goodness of fit statistics. Covariance structure models estimated in the present study allow non-recursive model estimations for the purposes of drawing inferences. These methods combined allow us to check the relationships using path analysis to simultaneously examine a number of correlated happiness, life satisfaction, and well-being variables. As reported earlier, most variables in the model were measured using a scale of (1 to 5); however, both life satisfaction and happiness used a scale of (0 to10). Values were standardized for further analysis. Table 2 provides the unstandardized values for the means and standard deviations of the variables.

## 4. Results

Table 3 provides a summary of the demographic profiles of the participants in the study. Overall, 61.6% of the sample were males and 79.9% were married. Approximately 44.3% were within the 35–44 age bracket, followed by 30.2% in the 24–34 age bracket. Moreover, 45.2% were holders of a bachelor’s degree. Emiratis constituted 41.3% of the sample, while non-Emiratis accounted for 58.7%.

The sample of respondents included different work categories (26.1% federal government, 33.1% local government, 14.5% semi government, and 26.3% private sector). Meanwhile, the working respondents covered a variety economic fields/activities of employers. The bulk came from education (20.7%), health and social work (13.3%), public administration (12.8%), defense (8.5%), financial and insurance (5.4%), water and electricity supply (4.3%), manufacturing (4.5%), transport and storage (4.2%), information and communication (4.7%), extra territorial organizations (4.1%), construction (4.9%), and other social and personal services (4.1%). The largest percentage of respondents came from the United Arab Emirates, India, Pakistan, Bangladesh, Yamen, Iraq, Afghanistan, Lebanon, Jordan, Syria, Iran, the Philippines, China, Japan, the UK, the USA, Germany, Italy, France, Spain, and Australia. Regarding monthly job income, 26.6% of the respondents made 25,000 Dirhams or less, 14.6% between 25,001 and 40,000 Dirhams, 6.8% between 40,001 and 60,000, 18.5% between 60,001 and 80,000 Dirhams, 13.5% between 8,.001 and 100,000, and 20% more than 100,000 Dirhams.

Table 4 shows the covariance matrix of the variables in the model. As suggested, the hypothesized model was examined via path analysis as a structural equation modeling method. Figure 1 shows the final path model and Table 5 presents the various model fit properties. The model demonstrated excellent fit (χ2/df = 1.7915 with a *p*-value of 0.27986, GFI = 0.99, AGFI = 0.98, CFI = 0.99, NFI = 0.98, NNFI = 0.98, SRMR= 0.000704, and RMSEA = 0.00139).

Table 6 shows the path estimates and their associated *t*-values for the variables in the path model. Life satisfaction is affected by five variables: job satisfaction, mental health, Social support network, satisfaction with relationships with others, and happiness. Nine variables influence happiness, including job satisfaction, mental health, Social support network, satisfaction with relationships with others, satisfaction with family life, self-assessment of health, quality time with family, work–life balance, and life satisfaction. The path model identifies direct reciprocal relations between life satisfaction and happiness for working people.

Table 7 focuses more on happiness and life satisfaction. It provides the direct, indirect, and total effects of the final variables in the path model. The direct and indirect effects of those well-being variables on happiness and life satisfaction can be revealed through a closer look at the figures. Concerning life satisfaction, mental health contributes the highest total effects (0.63554), followed by satisfaction with family life (0.56210), job satisfaction (0.50590), and Social support network (0.38096). Concerning happiness, the same order of well-being factors appears, as the highest total effects are related to mental health (0.84642), satisfaction with family life (0.82420), job satisfaction (0.63232), and Social support network (0.41448). It is worth noticing that the effect of job satisfaction is higher on happiness (0.63232) than on life satisfaction (0.50590).

Additional analysis was performed to better understand the presence of the recursive relation between life satisfaction and happiness. A related sensitivity analysis was conducted by controlling the directions of paths between happiness and life satisfaction. The model was run with only a path from happiness to life satisfaction, while the path from life satisfaction to happiness was removed. The model results are extensively different. The fit statistics are relatively poor, with mixed results (χ2/df = 789, GFI = 0.959, AGFI = 0.94, CFI = 0.962, NFI = 0.962, NNFI = 0.579, SRMR = 0.0283, and RMSEA = 0.169). The same poor results were obtained when the direction was set to be only from life satisfaction to happiness. Many other options were also experimented using the path model given the presence of other paths. The final model presented in Figure 1 provided the best fit.

## 5. Discussions

The well-being factors used in the path model replicated many other studies that adopted various analysis methods [23,33,46]. In general, the current study of working adults in Abu Dhabi provides evidence consistent with other international research that explains the influence of well-being factors on happiness or life satisfaction [43,67]. The results of the present study show that there is a statistically significant relationship between many well-being factors, happiness, and life satisfaction, while some factors play a significant role as a predictive factor of both happiness and life satisfaction.

As elaborated, while four factors specifically provided significant direct effects on life satisfaction and a higher number of well-being factors directly affected happiness, these variables exerted a total effect on either happiness or life satisfaction. The total effects were highly significant, irrespective of whether the effects were direct or indirect. This seems to confirm that certain well-being factors can have both direct effects and mediating effects by explaining the large variance of happiness or life satisfaction [68]. For example, work–life balance directly affects life satisfaction. At the same time, it also has an indirect influence on life satisfaction through the mediation of happiness. The same logic can be observed from three other variables imposing indirect effects on life satisfaction—satisfaction with family life, quality time spent with family, and self-assessment of health. Such a realization could mean that the use of the two concepts, happiness or life satisfaction, might reflect their similarity and cohesion as individuals look at life in its true meanings [69]. Moreover, the essence of the effect being direct or indirect may help to enhance our understanding when it comes to setting policies and strategies.

The constructs representing social connection, including satisfaction with family life, quality time spent with family, satisfaction with relations with other people, and how many people could support you, together provided a significant impact on both life satisfaction and happiness. Results support international research of the importance of the social connection of family and friends [24,25,26,27]. Such findings also generally support the assumption that there are adaptive relations between happiness or life satisfaction and social connection [1,70,71].

The main job- or income-related variable that significantly influences life satisfaction and happiness is job satisfaction, which shows both direct and indirect effects. This result is also consistent with those of many other well-being studies that show the significance of job satisfaction in people’s well-being [24,25,26,27]. Our understanding of such a relation supports the significance of work–life balance, which could be translated into contributing equal time to work and family attention, as referred to by others [72,73]. The implication for organizations is that following the principles of the science of happiness and offering more flexible work arrangements should be promoted at work to have happier employees [74]. The flexible work arrangement could give workers and employees greater scheduling freedom in how they fulfil their job responsibilities. Such strategies could better meet personal or family needs and achieve a better work–life balance. During the COVID-19 response timeframe, many workplaces provided a flexible work schedule to respond to school/daycare closures and other rapidly changing staffing needs specific to the circumstances. Indeed, in Abu Dhabi, many departments allowed a variety of flexible work arrangements that included reduced hours/part-time work, a compressed work week, telework/working remotely/telecommuting, flexible working hours, and job sharing.

Self-assessment of health is a significant contributor to happiness (directly) and life satisfaction (indirectly), offering support to other research that found strong correlations between happiness or life satisfaction and self-rated health [9,15,23,33]. Mental health alone also represented one of the significant contributors to both life satisfaction and happiness. Empirical research typically indicates the antagonistic relation between life satisfaction and negative mental feelings [75,76]. In this regard, this research lends support to the call for a more focused approach to life satisfaction and happiness with a clear understanding of the effects of positive mental health [49,50,51].

Strong evidence is presented in this research of life satisfaction and happiness playing major mediating roles. Many researchers have elaborated on the presence of direct, indirect, and mediation effects in quality-of-life studies [68,77]. The results of this study add to our understanding of the types of pathways that well-being factors travel to affect either the happiness or life satisfaction of working adults.

Many researchers have examined the relations between happiness and life satisfaction extensively and indicated a significant association between life satisfaction and happiness in many contexts [11,20,40]. The Abu Dhabi data show that there is a significant mutual influence between happiness and life satisfaction. Moreover, higher happiness results in better life satisfaction, and superior life satisfaction leads to a higher level of happiness. In short, this research reports and confirms the reciprocal or bidirectional relations between life satisfaction and happiness. Such a bidirectional relationship might provide and suggest that we be more careful about how the other variables investigated in the study might play a role. More analysis and understanding of related indicators are recommended. These might include socioeconomic status, such as income, race, and ethnicity, and type of employment, it is indeed highly beneficial to acknowledge that the bi-directional association between life satisfaction and happiness might be highly influenced by these socioeconomic factors. For example, life satisfaction and happiness might look significantly different for low-income and racial and ethnic minoritized populations than for high-income groups and other races. Proper interpretation of the non-recursive relationship between happiness and life satisfaction requires better and valid interpretations of how each variable is related to the other variables or categories of the community.

This study has some limitations to note. First, the sample over-represented working adults from the public sector compared to the private sector. For this reason, it is questionable whether the findings can be generalized to all working adults in Abu Dhabi.

## 6. Conclusions

The study provides insights into the reciprocal relationships between life satisfaction and happiness when examining well-being determinants for working adults in Abu Dhabi. It provides empirical evidence, perhaps for the first time, to show that the life satisfaction of working adults and their happiness mutually influence each other. Such high-level results suggest that working adults’ well-being factors and features may be dependent on the interconnection between their happiness and life satisfaction. In other words, life satisfaction plays a major role in happiness, and at the same time, feelings of happiness help working adults to enhance and record higher life satisfaction.

The study highlights the importance of relations with family and friends, job satisfaction, and mental health in affecting happiness and life satisfaction. It would be more constructive if the effects of these constructs were measured and studied over time. Furthermore, we confirm the importance and criticality of looking at happiness and life satisfaction more interactively with direct and indirect effects from many other well-being factors. In this sense, the findings of this research also provide practical guidance and evidence for policymakers to develop work systems that incorporate a broader vision of working people’s well-being by focusing on what is essential to their happiness and life satisfaction.

Future research should also look more closely at the effects of nationality, gender, age, marital status, living region in Abu Dhabi, and type of work to better understand life satisfaction and happiness. More specifically, future analysis could focus on the different populations in Abu Dhabi. Such analysis and focus could help us to understand any potential skews in the achieved sample. Such additions would, importantly, provide population reference material for future national and international comparisons. Because the study included working adults coming from different professions or types of work, future research could focus more on the differences in their life satisfaction and happiness by exploring these differences and how they could impact the interpretation of the related findings. The same observation could be raised regarding ethnicity and income.

## Figures and Tables

**Figure 1 ijerph-19-03575-f001:**
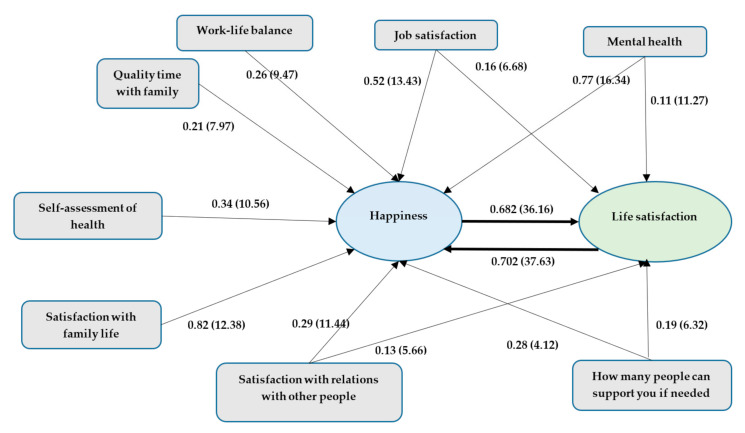
The path analysis model (estimate and *t*-value).

**Table 1 ijerph-19-03575-t001:** Scale values and explanations.

Variables in the Model and Explanations	Scale
Overall job satisfaction (JST): What is your overall job satisfaction?	(1: Very unsatisfied to 5: Very satisfied)
Work–family balance (WFB): How satisfied are you with the current balance between your job and home life?	(1: Very unsatisfied to 5: Very satisfied)
Self-perception of health (HS): In general, how do you assess your current health status?	(1: Very poor to 5: Excellent)
Mental health and depression (MN): During the past four weeks, how much of a problem did you have with feeling sad and depressed?	(1: Not at all, to 5: Very high). The results reversed for the analysis.
Social support network (SUP): How many people can help and support you whenever you need them?	(1: Not at all, to 5: A great extent)
Overall life satisfaction (LST): On a scale of 0–10, all things considered, how satisfied are you with your life nowadays?	(0: Extremely unsatisfied to 10: Extremely satisfied)
Satisfaction with relations with other people (SPR): How satisfied are you with the relationships with other people you know?	(1: Very unsatisfied to 5: Very satisfied)
Feelings about current surrounding environment (ENV): How do you feel about your current surrounding living environment?	(1: Very bad to 5: Very good)
Amount of quality family time (FM1): How would you describe the amount of quality time you spend with your family?	(1: Very limited quality time to 5: Much quality time)
Satisfaction with family life (FM2): In general, I am satisfied with my family life	(1: Strongly disagree to 5: Strongly agree)
Often socially meeting with friends (FRD): How often do you meet socially with friends?	(1: No recent contacts to 5: Several times a week)
Happiness (HPY): On a scale of 0 to 10, how would you describe your average level of happiness as an Abu Dhabi resident?	(0: Extremely unhappy to 10: Extremely happy)

**Table 2 ijerph-19-03575-t002:** Scale types, means, and standard deviations of variables in the model.

	Variables in the Model	Mean	Standard Deviation
JST	Overall job satisfaction	3.489	1.099
WFB	Work–family balance	2.996	1.132
HS	Self-perception of health	3.139	1.014
MN	Mental health and depression	3.189	1.03034
SUP	Social support network	2.701	1.217
LST	Overall life satisfaction	6.551	2.369
SPR	Satisfaction with relations with other people	3.802	.831
ENV	Feelings about current surrounding environment	3.626	1.032
FM1	Amount of quality family time	2.650	1.206
FM2	Satisfaction with family life	3.895	1.091
FRD	Often socially meeting with friends	2.224	1.204
HPY	Happiness	6.972	2.437

**Table 3 ijerph-19-03575-t003:** Demographics of the participants.

	Number	Percentage
Sex		
Male	21,083	61.1%
Female	13,416	38.9%
Marital status		
Married	27,573	79.9%
Single	5300	15.4%
Divorced	1203	3.5%
Separated	251	0.7%
Widowed	172	0.5%
Education level		
Illiterate	63	0.2%
Below secondary school	1128	3.3%
Secondary school	4811	13.9%
Post high school training certificate	1397	4.0%
College diploma	3761	10.9%
Bachelor’s degree	15,605	45.2%
Master’s degree	6762	19.6%
Doctorate degree	972	2.8%
Age		
24 or less	1339	3.9%
25–34	10,412	30.2%
35–44	15,294	44.3%
45–54	6151	17.8%
55–64	1303	3.8%
Nationality		
Emirati	14,247	41.3%
Non-Emirati	20,252	58.7%

**Table 4 ijerph-19-03575-t004:** The covariance matrix of variables in the model.

	JST	WFB	HSP	MNT	SUP	FRD	FM1	FM2	SPR	ENV	LST	HPY
JST	0.972											
WFB	0.356	0.978										
HSP	0.214	0.297	0.950									
MNT	0.251	0.395	0.284	0.918								
SUP	0.104	0.148	0.085	0.103	0.133							
FRD	0.052	0.041	0.070	0.111	0.165	0.118						
FM1	0.134	0.151	0.132	0.364	0.171	0.299	0.893					
FM2	0.212	0.552	0.251	0.390	0.261	0.325	0.387	0.877				
SPR	0.135	0.148	0.205	0.267	0.221	0.258	0.198	0.186	0.965			
ENV	0.299	0.208	0.218	0.224	0.158	0.188	0.259	0.225	0.390	0.945		
LST	0.314	0.429	0.338	0.343	0.274	0.392	0.377	0.264	0.247	0.295	0.881	
HPY	0.284	0.337	0.337	0.317	0.248	0.320	0.349	0.276	0.255	0.337	0.623	0.894

**Table 5 ijerph-19-03575-t005:** Goodness of fit statistics for the final model.

Fit Statistics and Properties	Values
Degrees of Freedom	4
Maximum Likelihood Ratio Chi-Square	7.166
Root Mean Square Error of Approximation (RMSEA)	0.002
*p*-Value for Test of Close Fit	0.279
Normed Fit Index (NFI)	0.990
Non-Normed Fit Index (NNFI)	0.981
Comparative Fit Index (CFI)	0.991
Goodness of Fit Index (GFI)	0.990
Adjusted Goodness of Fit Index (AGFI)	0.982
Root Mean Square Residual (RMR)	0.028
Standardized RMR	0.001
Parsimony Goodness of Fit Index (PGFI)	0.046

**Table 6 ijerph-19-03575-t006:** Path analysis model—specifications.

Life Satisfaction (Path From)	Estimates	*t*-Values
Job satisfaction	0.16	6.68
Mental health	0.11	11.6
Social support network	0.19	6.32
Satisfaction with relations with others	0.13	5.66
Happiness	0.682	36.16
**Happiness (Path from)**	**Estimates**	***t*-Values**
Job satisfaction	0.52	14.43
Mental health	0.77	16.34
Social support network	0.28	4.12
Satisfaction with relations with others	0.29	11.44
Satisfaction with family life	0.82	12.38
Self-assessment of health	0.34	10.56
Quality time with family	0.21	7.97
Work–life balance	0.26	9.47
Life satisfaction	0.702	37.63

**Table 7 ijerph-19-03575-t007:** Direct and indirect effects in the reciprocal model.

From	To	Direct Effect	Indirect Effect	Total Effect
Work–life balance	Life satisfaction	-----	0.177	0.177
Work–life balance	Happiness	0.264	-----	0.264
Job satisfaction	Life satisfaction	0.160	0.355	0.506
Job satisfaction	Happiness	0.520	0.112	0.632
Mental health	Life satisfaction	0.110	0.525	0.636
Mental health	Happiness	0.772	0.075	0.847
Social support network	Life satisfaction	0.190	0.191	0.381
Social support network	Happiness	0.281	0.134	0.415
Satisfaction with relations with other people	Life satisfaction	0.132	0.198	0.330
Satisfaction with relations with other people	Happiness	0.293	0.091	0.384
Satisfaction with family life	Life satisfaction	------	0.562	0.562
Satisfaction with family life	Happiness	0.824	-----	0.824
Quality time spent with family	Life satisfaction	------	0.146	0.146
Quality time spent with family	Happiness	0.215	-----	0.214
Self-assessment of health	Life satisfaction	------	0.234	0.234
Self-assessment of health	Happiness	0.343	-----	0.340
Happiness	Life satisfaction	0.682	-----	0.680
Life satisfaction	Happiness	0.702	-----	0.702

## Data Availability

The data presented in this study are available on request from the corresponding author. The data are not publicly available due to privacy restrictions.

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
