# Peer review of "Exploring the Reciprocal Relationships between Happiness and Life Satisfaction of Working Adults—Evidence from Abu Dhabi"

_ijerph, 2022, doi:10.3390/ijerph19063575_

Round 1
Reviewer 1 Report
The article is devoted to an interesting and important problem. The Introduction section accurately describes the contents of the subsequent manuscript. The Literature Review section contains references to studies relevant to the research objectives presented by the authors. The Methodology and Results of the study are described quite clearly. In the Discussion, the authors compare their results with the results of similar studies and include them in the modern context of investigating predictors of happiness and well-being. In my opinion, the most vulnerable point in the manuscript is its title. The title includes words "temporal dimensions", but there is no information about temporal dimensions either in the abstract, in the text of the manuscript, or in the conclusions. It is unclear what the authors meant by "temporal dimensions", since the procedure for their assessment is also not described in the Methodology section. In my opinion, the title of the manuscript should be corrected in order to more accurately reflect the results that are presented in the text, or the authors should clarify the answers to these questions in the text of the manuscript. After that, the article can be recommended for publication.
Reviewer 2 Report
For the uninitiated, the concepts of wellbeing, life satisfaction, happiness and quality of life are not well differentiated. Indeed, it is commonplace to find in the literature that some of these terms are used interchangeably. Within this paper, life satisfaction and happiness are described as “two subjective measures of well-being”. It would be helpful to include some definitions and/or clarification of these various terms, partly to enable the reader to understand the conceptual landscape but also to help clarify the notion of differences in domain “size”, as suggested when comparing life satisfaction and happiness.
Complexity here is further increased in differentiating between so-called objective / subjective measures. If we accept that satisfaction/happiness are, by definition, subjective constructs, then it is important to understand how any metric can be other than subjective in its nature. The situation is different were we to record objective/subjective responses to a quantifiable entity, for example measuring actual/perceived weight of an object.
This paper reports quantitative results based on a large population survey. It is wholly necessary in such a report to fully describe the key variables. Table 2 is completely lacking in meaning without knowing specifically, the question that was presented in the survey instrument. As it stands a reader can make little or no sense of a self-perception of health “score” of 3.139 on an invisible 5-point rating scale. It is imperative that a full description is provided of the survey questions used to derive the key variables in this study. Without this it is impossible to pass judgment on the results that are presented here.
Two other aspects of the pre-processing of survey data should be noted. Firstly, the level of measurement needs to be correctly identified. It is not immediately obvious why a 5-point rating scale that generates categorical responses should be treated as a cardinal variable and reported in terms of a mean and SD. Secondly, the need for standardisation of variable scores needs to be accounted for – why was this necessary and how was it achieved.
As reported elsewhere, the survey was launched in both English and Arabic language versions. This important distinction seems to have been overlooked in this current study. Language preference is of course likely also to be connected with ethnicity and/or country of origin. Web-based sources suggest that Indian and Pakistani adults account for ~40% of the total population. Ignoring language preference is a non-starter in reporting such survey data. At the very least there needs to be some evidence that this is NOT a significant obstacle to aggregating the data from what are effectively two different versions of the survey.
It would be interesting to have ethnicity, or failing this then language preference, included in the substantive analysis. Differential acculturation might influence the relationships between variables of interest. For example, Arab-speaking Indians might be thought of as being better assimilated, with consequentially higher levels of happiness and satisfaction when compared with English-speaking Indians (other things being equal).
Table 1 could be usefully amended to include corresponding statistics for the UAE population. This would help understand any potential skews in the achieved sample but would importantly, provide population reference material for future national and international comparisons.
On a more detailed point, it seems excessive to report data values to more than 3 places of decimal. Please review the Tables accordingly.
It is not possible to comment further on this paper without a proper understanding of the source data including descriptive analysis of the key variables.
Reviewer 3 Report
File attached.

Round 2
Reviewer 3 Report
See attached file.
